# Exploring Blockchain Research in Supply Chain Management: A Latent Dirichlet Allocation-Driven Systematic Review

Abderahman Rejeb [1,*], Karim Rejeb [2], Steve Simske [3] and John G. Keogh [4]

1  Department of Management and Law, Faculty of Economics, University of Rome Tor Vergata, Via Columbia, 2, 00133 Rome, Italy
2  Faculty of Sciences of Bizerte, University of Carthage, Bizerte 7021, Tunisia; karim.rejeb@fsb.ucar.tn
3  Systems Engineering Department, Colorado State University, Fort Collins, CO 80523, USA; steve.simske@colostate.edu
4  McGill Centre for the Convergence of Health and Economics, McGill University, 680 Sherbrooke, St. W., Montreal, QC H3A 0B8, Canada; john.keogh@mcgill.ca
*  Correspondence: rjbbrh01@uniroma2.it

**Abstract:** Blockchain technology has emerged as a tool with the potential to enhance transparency, trust, security, and decentralization in supply chain management (SCM). This study presents a comprehensive review of the interplay between blockchain technology and SCM. By analyzing an extensive dataset of 943 articles, our exploration utilizes the Latent Dirichlet Allocation (LDA) method to delve deep into the thematic structure of the discourse. This investigation revealed ten central topics ranging from blockchain's transformative role in supply chain finance and e-commerce operations to its application in specialized areas, such as the halal food supply chain and humanitarian contexts. Particularly pronounced were discussions on the challenges and transformations of blockchain integration in supply chains and its impact on pricing strategies and decision-making. Visualization tools, including PyLDAvis, further illuminated the interconnectedness of these themes, highlighting the intertwined nature of blockchain adoption challenges with aspects such as traceability and pricing. Despite the breadth of topics covered, the paper acknowledges its limitations due to the fast-evolving nature of blockchain developments during and after our analysis period. Ultimately, this review provides a holistic academic snapshot, emphasizing both well-developed and nascent research areas and guiding future research in the evolving domain of blockchain in SCM.

**Keywords:** blockchain technology; supply chain management; traceability; supply chain finance; e-commerce; topic modeling



## 1. Introduction

Supply chains have undergone remarkable transformations recently, transitioning from a mere operational aspect to a stand-alone supply chain management (SCM) role [1]. These chains encompass various strategic and logistical tasks, such as strategizing, executing, and supervising the efficient movement and storage of products, services, and associated information from the origin to the end user to meet their needs [2,3]. Enhancing and consolidating these tasks can lead to enhanced transparency, revenue growth, faster inventory cycles, efficient movement of goods, and proficient customer service [4,5]. However, realizing these goals is a daunting task due to the intricacies of supply chains, which have grown exponentially and become more complex. This is primarily because of the involvement of multiple supply sources spread across different geographic regions, each working autonomously and often in competition to cater to their client's needs [6–8]. In addition to these intricacies, supply chains face a myriad of challenges, heightened by the inherent complexities and evolving landscape of global trade. Prominent among these challenges are the unexpected disruptions caused by natural disasters (e.g., the Yogyakarta earthquake in Japan) [9] or geopolitical tensions (e.g., Russia–Ukraine conflict) [10], which

can strain or even sever crucial supply links. Dishonest actions by trading allies, such as misinformation or deceit [11], add complexity. Breaches in data privacy [12] pose significant risks, particularly in an era of increasing digital connectivity. Many sectors of industry grapple with an increased risk of digital crimes (e.g., hacking to steal intellectual property and ransomware attacks) and the resource-intensive task of market surveillance for counterfeit products and supplier monitoring to identify sub-standard components or ingredients [13]. Recent global events, such as the economic conflict that ignited a trade war between the USA and China in 2018, and the COVID-19 pandemic, have further underscored the need for resilience and adaptability in SCM [14,15].

Recently, many companies have started to pilot and adopt blockchain technology to enhance their operations and better oversee their supply chains [16–18]. Essentially, blockchain is a "digital, distributed ledger that chronologically logs transactions, aiming for enduring and unalterable records" [19] (p. 547). This distributed record comprises chronologically arranged blocks secured through cryptographic techniques [20]. Each block holds specific entries, such as data or records, which are integrated into the network and are connected to the previous block. Once integrated into the blockchain, these blocks cannot be altered and are authenticated through intricate automation and procedures [21]. Built on peer-to-peer (P2P) networks, blockchain requires consensus among participants to authenticate transactions, enabling the removal of false or deceitful records. Distinct from traditional IT systems, blockchain reduces the dependence on a central authority, promoting secure and quasi-anonymous dealings among parties [22,23]. Per Rejeb et al. [24], an individual blockchain solution is tailored to tackle specific challenges or business scenarios, indicating its adaptability across various sectors. The pseudonymous Satoshi Nakamoto [25] introduced blockchain in 2008 with Bitcoin, marking an innovative strategy for creating trust-based systems. In the financial domain, blockchain's secure nature prevents 'double spending,' guaranteeing each transaction's validity without duplication [26]. Apart from its role in finance, blockchain's utility spans sectors, such as logistics and SCM [27–29], social media [28,30], online retail [31,32], travel [33,34], and healthcare [35–37]. Correspondingly, blockchain research, especially within logistics and SCM, has surged [38].

The heightened interest in blockchain research stems from various factors. Firstly, blockchain introduces novel paradigms for business structures, management models, and organizational systems, enhancing resource allocation, traceability, trust, and data security [39,40]. For example, Gayialis et al. [41] present a comprehensive reference model for wine traceability, emphasizing the potential of blockchain-enabled systems that integrate value chains, organizational resources, business functions, and risks to bridge the gap between developers and stakeholders.

Khanna et al. [42] propose a blockchain-based platform for India's dairy industry, utilizing smart contracts, QR codes, and IoT to address concerns about dairy-related food fraud and elevate social, economic, and sustainability standards. Kechagias et al. [43] introduce a distributed application for table olives' traceability on the Ethereum network. The project showcased a successful implementation by a Greek producer by enhancing supply chain efficiency, data reliability, and regulatory compliance. Wang et al. [44] present a combined blockchain and RFID-based traceability system, embedding blockchain data within RFID tags for decentralized, battery-free product traceability, suitable for industrial scalability. Beyond the food industry, Omar et al. [45] introduce a blockchain solution using Ethereum smart contracts for personal protective equipment (PPE) supply chain challenges during COVID-19, enhancing transparency and security. Collectively, these selected studies underscore the diverse and transformative applications of blockchain technology across various industries, highlighting its potential to enhance traceability, transparency, trust, and operational efficiency in SCM.

Furthermore, blockchain can help to reshape transaction and relationship dynamics among supply chain stakeholders [17] by enhancing trust, enabling integration, and reinforcing collaboration [46]. Importantly, blockchain can help to modernize key aspects of

logistics and SCM operations by advancing sustainable practices and optimizing processes such as distribution and information exchange [47].

Blockchain's application in SCM has increasingly piqued academic and industrial interest, as reflected in the plethora of reviews that have been conducted. Gurtu and Johny [48] embark on a systematic review of the literature and posit the formidable potential of blockchain in SCM, emphasizing its capability to enhance efficiency by eliminating intermediaries. Their review is based on an analysis of 299 papers from the EBSCO database up to December 2018. Similarly, Queiroz et al. [49] analyze 27 articles spanning 2008 to 2018 and underscore the nascent yet promising nature of blockchain–SCM integration, with some sectors, such as the electric power industry, already exhibiting mature blockchain applications, notably smart contracts. Through a systematic literature review of 37 publications, Varriale et al. [50] delineate the sustainable implications of blockchain in supply chains, revealing its multi-dimensional impact on environmental, economic, and social facets while underscoring both its benefits for business profitability and reputation, as well as the extant research gaps for future exploration. Wang et al. [51] review the transformative potential of blockchain in supply chains, identifying its value in enhancing visibility, traceability, and digitalization, and underscore both its challenges and broader socio-economic implications. Hastig and Sodhi [52] examine supply chain traceability in industries such as cobalt mining and pharmaceuticals, identifying core business requirements and critical factors for successful blockchain implementation.

Musigmann et al. [53] investigate the impending ramifications of blockchain on supply chain practices and policies. The authors conclude that trust is blockchain's main feature as it gains traction in SCM. The research highlights potential applications that encompass enhanced transparency, digital transformation, reinforced data security, and the inception of smart contracts. Rejeb et al. [38] and Pournader et al. [54] adopt a bibliometric approach to explore the blockchain in logistics and SCM terrain. While Rejeb et al. [38] classify 613 articles into meaningful clusters, including theoretical frameworks and practical applications, the study by Musigmann et al. [53] scrutinizes 628 papers from 2016–2020 and emphasizes the academic structures, seminal works, and pioneering scholars in this space. It unveils themes centered around blockchain's conceptualization, potentialities, adoption challenges, and roles in fortifying supply chain agility and intellectual property protection.

Pournader et al. [54] research blockchain's implications in supply chains, logistics, and transport and present four pivotal co-citation clusters: Technology, Trust, Trade, and Traceability/Transparency, each representing emerging themes for blockchain in SCM. Furthermore, Van Nguyen et al. [55] utilize a distinctive methodology that leveraged the power of Latent Dirichlet Allocation (LDA) combined with text mining for autonomous content analysis of full-text articles. Their subsequent analysis of 108 articles from 2017–2022 identifies ten salient research themes, including revenue management, decentralized autonomous organizations in SCM, and more. The primary objectives of their study are to provide a comprehensive review of the literature on the subject matter, identify emerging trends, and present novel analytical methods to further enhance our understanding of the blockchain-supply chain nexus. Meanwhile, while our study extracts valuable insights from these research findings, it carves a distinct niche in several dimensions. First, our sample encompasses a broader dataset beyond the boundaries set by previous systematic literature reviews [51,55]. Second, in a strategic departure from the full-text-centric approach of earlier research, we focused our attention on article abstracts. This refined focus facilitates a more agile yet incisive examination, adeptly capturing the crux of each document without delving into its main body.

Methodologically speaking, while we continue to rely on the robust foundation of LDA, we introduce a pivotal modification: Our algorithm is meticulously calibrated to discern patterns, trends, and subtle undertones within abstracts. This optimization ensures that our process is proficient in extracting valuable insights from brief summaries. This fusion of traditional bibliometric techniques with a machine learning-enhanced LDA framework

introduces a renewed analytical paradigm, extending the traditional boundaries of what systematic reviews can achieve.

Our review's contributions are multifaceted. Our emphasis on abstracts presents a methodology of analysis that is scalable and efficient, hence expanding the research capabilities to include comprehensive and exhaustive syntheses in future meta-studies. The expansive nature of our dataset review offers a panoramic view, incorporating a plethora of perspectives and insights. Additionally, the blending of a refined LDA approach with bibliometric techniques contributes to methodological innovation and the advancement of systematic review methodologies.

The organization of this article is as follows: Section 2 articulates the research methodology, and Section 3 presents a concise summary of the review's primary findings. Section 4 delves deeper into the themes discovered, and the article concludes with overarching reflections and a brief discussion of the inherent limitations of our review.

## 2. Methodology

For our study, we utilized the LDA technique, a multi-layered Bayesian model rooted in three primary levels [56]. Essentially, LDA is anchored on algorithms tailored to pinpoint and categorize underlying topics within voluminous text collections. This method operates under the assumption that each document in the dataset encompasses various themes, with each theme characterized by a unique word composition [57]. LDA's primary objective is to explore these latent themes by assessing patterns of word occurrences and their interplay across documents. Using LDA, scholars can delineate K topics, denoted as βK, representing the likelihood distributions over a corpus of terms within the vocabulary V. One of the notable strengths of the LDA approach is its proficiency in autonomously recognizing and extracting relevant patterns from expansive text repositories. This compatibility and efficiency made LDA particularly suitable for our exploration, allowing for an exhaustive and unbiased assessment of an extensive body of literature, drawing inspiration from earlier successful applications [58,59]. For statistical analysis, visualization, and natural language processing (NLP), we harnessed the capabilities of both R and Python programming tools.

### 2.1. Data Collection

In July 2023, we began to systematically collate publications that combine blockchain technology with SCM. Utilizing the esteemed Web of Science database, our search parameters were tailored to prioritize articles and review articles in English. To refine our search further, we selected the following categories: Management, Operations Research Management Science, Business, Economics, Business Finance, Transportation, Transportation Science Technology, and Social Sciences Interdisciplinary. We excluded conference proceedings papers and book chapters from our search. This strategy provided a dataset of 1013 potentially relevant research articles.

We incorporated insights from the Scopus database, a repository held in high regard for its encyclopedic compilation of academic literature [60]. Our investigation on Scopus revolved around a strategic selection of keywords geared to encapsulate the multifaceted domain of blockchain in SCM. Our query design was expansive, ensuring we did not overlook any blockchain-related terminology pivotal to the supply chain context. The search query was as follows: blockchain AND ("supply chain*" OR logistic*) [38]. By employing the asterisk (*) as a wildcard, we maximized the scope of our search, capturing varied iterations of focal keywords. We selected English language articles, a convention underscored by best research practices [61]. After anchoring our search in journal articles only, we then embarked on a thorough search of the blockchain-supply chain nexus, aiming to unravel the depth of extant knowledge while highlighting emergent research themes [62]. After the initial assessment of the dataset and after eliminating duplicates and redundant articles, two independent authors conducted a thorough review of the abstracts to negate any biases. This dual-review process culminated in the shortlisting of 943 articles deemed most pertinent to our topic, setting the stage for our final analysis. The selected papers

were published within a research time horizon spanning from 2017 to July 2023, providing a comprehensive overview for our final analysis. For a detailed visual representation of our selection procedure, please refer to the PRISMA flowchart in Figure 1.

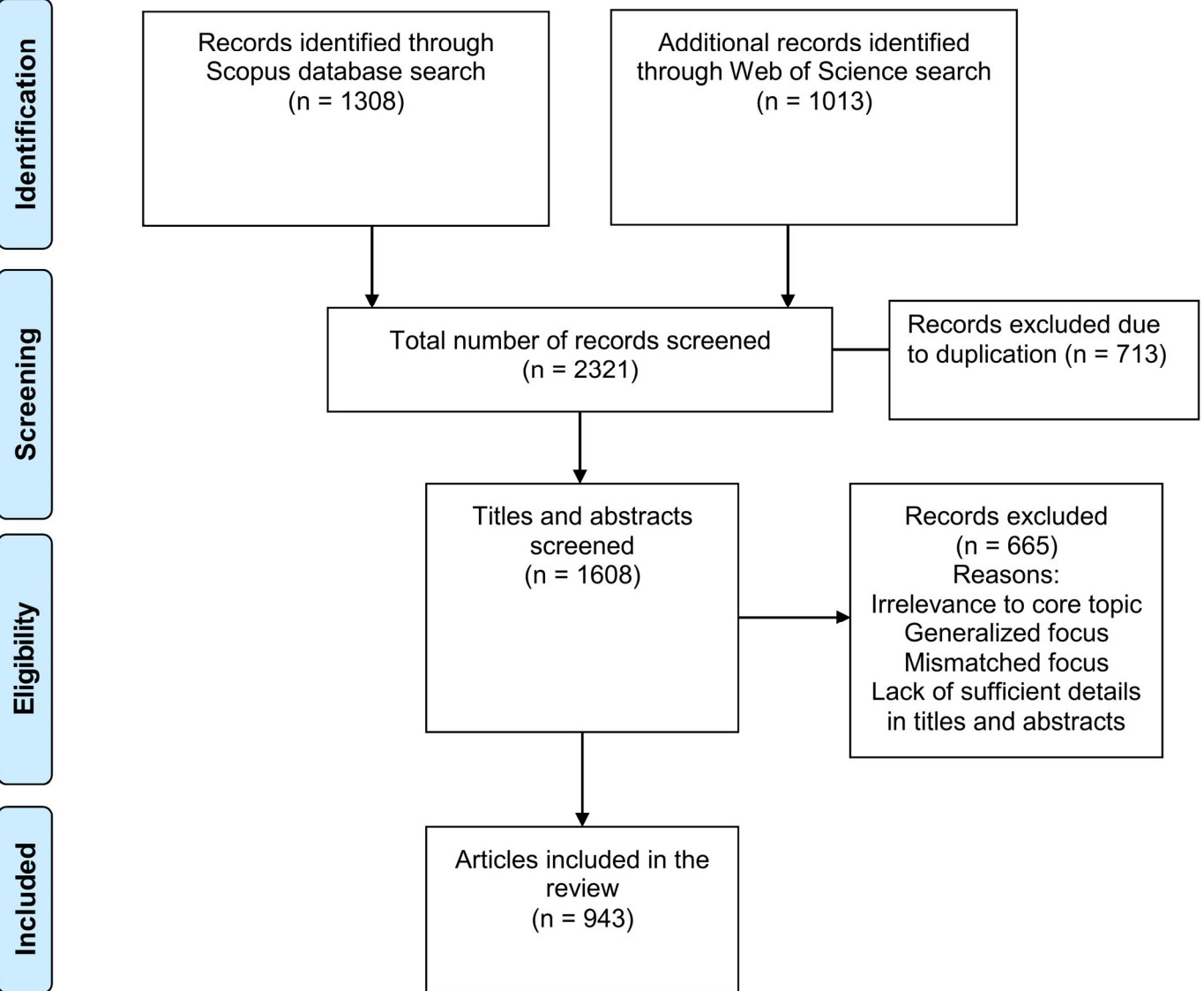

**Figure 1.** Detailed PRISMA Flowchart illustrating the systematic screening and selection process for articles on blockchain research in SCM.

We utilized text-mining techniques and tools aimed at extracting insights from the dataset of abstracts. In doing so, we replicated methodologies from previous LDA-based investigations [63]. LDA, with its probabilistic topic model approach, adeptly finds topics based on the co-occurrence of words, making it particularly suited for our analysis of diverse abstracts in the SCM domain. Our focus on abstract-level content is aimed at identifying the applications of blockchain in SCM. Hence, our analysis provides a holistic view of blockchain's multifaceted implications, opportunities, and challenges in reshaping supply chain dynamics.

### 2.2. Textual Data Preparation for LDA Analysis

Before deploying the LDA technique, it was imperative to refine and pre-process the textual dataset associated with blockchain in SCM. We started this by removing extraneous elements such as line breaks, web links, punctuation marks, and other unrelated symbols. Leveraging the capabilities of Python's Gensim library, we further streamlined the content

by filtering out verbs, adverbs, adjectives, and common stop words. To align the analysis more closely with our study's context, we augmented Gensim's default stop word list with specific terms considered irrelevant to blockchain's integration within supply chain frameworks. Towards the end of this dataset cleansing process, Gensim played a pivotal role in breaking down the text into individual word tokens and assigning them unique identifiers. This approach facilitated an understanding of word prevalence and its significance within the domain of blockchain in supply chain research.

### 2.3. Formulating the LDA Model and Establishing the Ideal Topic Quantity

Establishing a dataset vocabulary is a pivotal step when crafting an LDA model to extract themes from textual data. Using Gensim's 'id2word' function, we transformed the words into a vectorized format. Subsequently, we utilized the Mallet toolkit for its proficiency in topic modeling, document categorization, and data clustering [64]. During this modeling phase, numerous parameter calibrations were required. As such, Mallet was tasked to perform multiple LDA iterations, each spotlighting a varied topic quantity.

To pinpoint the optimal topic number, we relied on the coherence score. This score offers an evaluation metric that gauges topic cohesiveness by analyzing inter-word relationships within each topic. At its core, the coherence score assigns a quantitative value to every topic, mirroring the interlinkages among its constituent words. Typically, a greater coherence score signifies a stronger bond among words within a topic, thus ensuring a clearer thematic delineation crucial for proficient topic modeling. Figure 2 presents the coherence scores that emerge from this methodology. Using coherence metrics as a guide, the most advantageous LDA model is consistently the one with the highest score, indicating superior topic clarity. With 10 topics in the model, the coherence score reached its highest point of 0.38051. As seen in Table 1, additional topics beyond this count would not substantially amplify the thematic precision. In light of the coherence values extracted from the LDA iterations, we inferred that the model centered on 10 topics resonated best with our analytical objectives. Table 1 catalogs the coherence scores tied to the different topic numbers we examined.

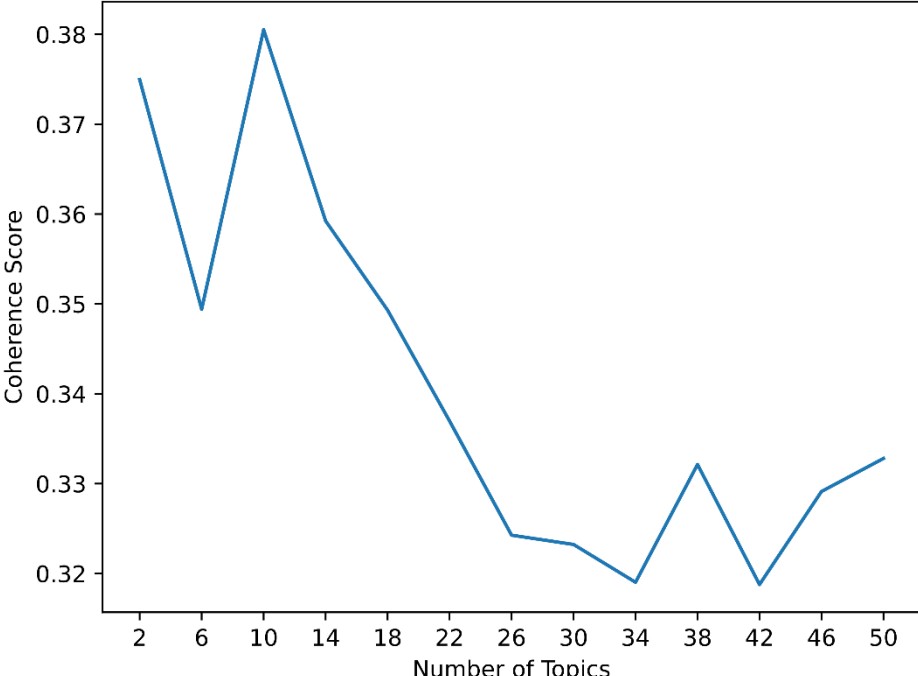

**Figure 2.** Coherence scores plot.

**Table 1.** Coherence scores value.

| Number of Topics | Coherence Score |
| --- | --- |
| 2 | 0.37496 |
| 6 | 0.34941 |
| 10 | 0.38051 |
| 14 | 0.35924 |
| 18 | 0.34930 |
| 22 | 0.33696 |
| 26 | 0.32426 |
| 30 | 0.32323 |
| 34 | 0.31902 |
| 38 | 0.33213 |
| 42 | 0.31877 |
| 46 | 0.32913 |
| 50 | 0.33280 |

### 2.4. Unveiling Themes: The LDA Procedure

The LDA algorithm functions as a statistical model for identifying underlying themes within a collection of documents [65]. In our study, the focus was on research centering around blockchain's application in SCM. In the graphical representation shown in Figure 3, the rectangles act as iterative indicators. Herein, 'M' denotes the documents, with 'N' signifying the topic occurrence within them. The words, symbolized by 'w', stem from the topic allocation represented by 'z'. Within this schema, 'β' underscores the distribution of words across topics, 'θ' portrays the topic distribution across documents, and 'α' indicates the word distribution within specific topics.

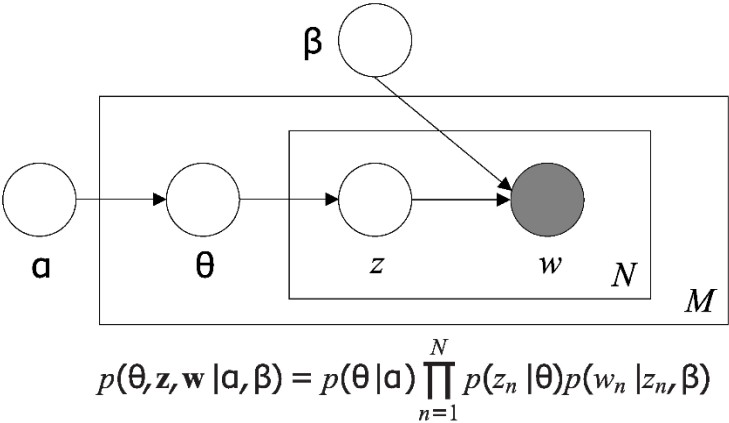

$$p(\theta, \mathbf{z}, \mathbf{w} \mid \alpha, \beta) = p(\theta \mid \alpha) \prod_{n=1}^{N} p(z_n \mid \theta) p(w_n \mid z_n, \beta)$$

**Figure 3.** LDA model representation [65].

By using the LDA algorithm, we found the frequency of the ten topics from our literature exploration. The semantic coherence metric was applied to find the prominence of terms linked to each theme in the article summaries. Two researchers worked independently and adhered to a methodology based on the semantic coherence scores to compile articles that correspond to each theme. This approach reinforced the LDA model's capability to pinpoint the inherent themes in each paper, outlining both their relevance and periodicity within the content. This aided in identifying the dominant research trajectory for blockchain in the SCM domain.

To undertake the LDA analysis on the abstracts dataset, a number of Python libraries were used. For example, PyLDAvis was used to discern the average distance between

topics and to spotlight the top ten terms from our dataset. Additionally, the Matplotlib toolkit was used to graphically represent our findings. These tools helped to improve the clarity and interpretative depth of the findings.

### 2.5. Deep Dive into Scholarly Insights: Bibliometric Exploration

To understand the academic contributions of blockchain in SCM, we completed a bibliometric examination using the R software package [66]. The software enables scholars to comprehend the inherent networks and dominant narratives within a dataset by facilitating the identification of interconnections among publications.

Our primary investigative objectives included a comprehensive evaluation and the creation of a scientific visualization. These objectives were attained through bibliometric techniques [67] that provided an in-depth analysis of academic collaboration, scholarly outputs, and insights into the evolution of blockchain in SCM.

## 3. Findings

A comprehensive bibliometric analysis was conducted in order to decipher the academic discourse surrounding blockchain research in SCM. The analysis performed on the dataset is summarized in Table 2. Notably, the publications in the dataset have averaged 1.38 years in circulation and 35.23 citations. This highlights the nascent nature of the domain and the burgeoning scholarly interest in the intersection of blockchain and SCM.

**Table 2.** Main bibliometric results.

| Description | Results |
| --- | --- |
| Main information about the data | |
| Timespan | 2017:2023 |
| Sources (Journals, Books, etc.) | 253 |
| Documents | 943 |
| Average years from publication | 1.38 |
| Average citations per document | 35.23 |
| Average citations per year per doc | 10.63 |
| References | 58,264 |
| Document types | |
| article | 868 |
| article; early access | 8 |
| review | 65 |
| review; early access | 2 |
| Document contents | |
| Keywords Plus (ID) | 2497 |
| Author's Keywords (DE) | 2165 |
| Authors | |
| Authors | 2409 |
| Author Appearances | 3317 |
| Authors of single-authored documents | 46 |
| Authors of multi-authored documents | 2363 |
| Authors collaboration | |
| Single-authored documents | 58 |
| Documents per Author | 0.391 |
| Authors per Document | 2.55 |
| Co-Authors per Documents | 3.52 |
| Collaboration Index | 2.67 |

The collaboration dynamics within the research community offer additional insights. For example, 46 publications had a single author, whereas 2363 publications had multiple co-authors with an overall Collaboration Index of 2.67. The latter statistic might be indicative of an interdisciplinary approach to researching the role of blockchain in the supply chain domain.

By analyzing the data from 2017 to July 2023 (as illustrated in Figure 4), multiple phases of academic research can be identified. In 2017, three articles were published, marking the beginning of research into blockchain in SCM. In 2018, seventeen articles were published, followed by forty-five in 2019. The data show a significant increase in publications from 2020, with 122 to 279 in 2022, indicative of the burgeoning recognition of the synergies between blockchain and supply chain systems. This growth can be perceived as a reflection of the global paradigm shift towards decentralized, transparent, and secure supply chain systems powered by blockchain. As of our cut-off date in July 2023, the research momentum in this domain continues with 292 research publications and is on pace to exceed 400 publications for the full year.

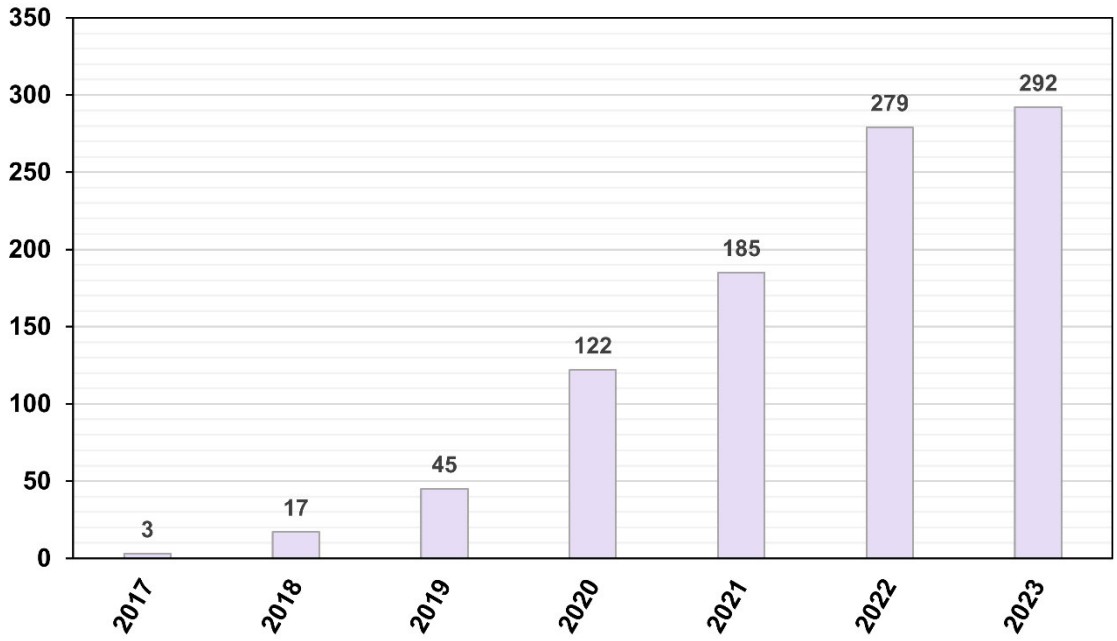

**Figure 4.** Year-wise distribution of selected journal articles.

Figure 5 highlights the significance of certain journals within the domain of blockchain research in SCM. For example, Sustainability is a leading journal focusing on holistic and responsible approaches and underscoring the alignment of blockchain solutions with sustainable SCM. The International Journal of Production Research highlights the integration of blockchain technology in optimizing and streamlining production processes.

Furthermore, Transportation Research Part E: Logistics and Transportation Review showcases the growing recognition of blockchain's potential in transportation and logistics solutions, suggesting the decentralizing shift in these domains. The prominence of the Annals of Operations Research and the International Journal of Production Economics reinforces the interconnectedness of blockchain, operations, and economic considerations in the supply chain context.

These journals collectively portray a rich and multi-dimensional academic landscape for blockchain in SCM. Their varied focuses—spanning sustainability, production, transportation, operations, and economics—manifest the cross-disciplinary character of blockchain research. This suggests that while blockchain offers transformative solutions in SCM, it indeed requires a blend of technological, operational, and economic insights.

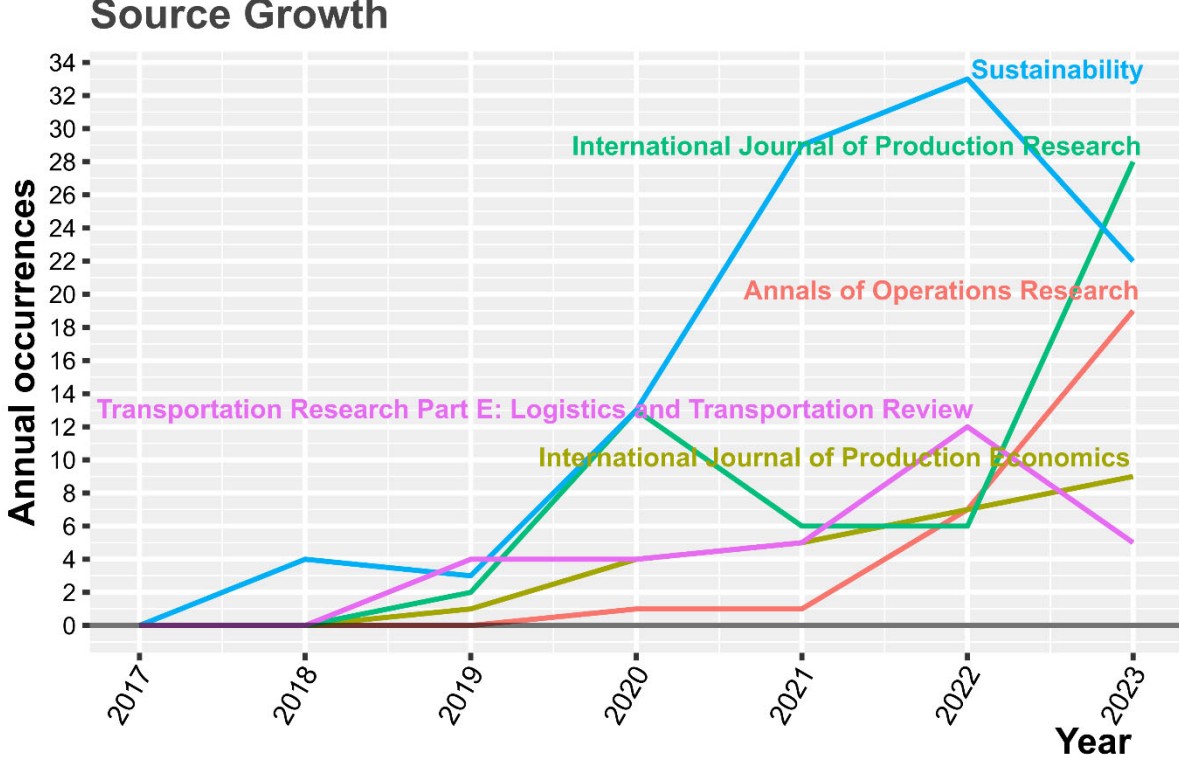

**Figure 5.** Most productive journals in the field of blockchain research in SCM.

The main goal of this study is to design and utilize an LDA model that analyses the scholarly discourse of blockchain in SCM. The researchers uncovered ten pivotal terms and their corresponding significance measures. Through inductive evaluation, patterns linked to these themes were identified and provided a thorough assessment of the prevalent research on blockchain applications in SCM.

Within the extensive body of literature under examination, two topics emerge prominently as subjects of keen debate: the challenges, transformations, and adoption processes associated with blockchain in SCM (Topic 8) and the effects of blockchain on pricing strategies and decision-making within the same domain (Topic 2). The intensity of the discourse surrounding these topics may suggest that these areas are undergoing rapid evolution, with varied perspectives and opinions among researchers and experts. Such fervent debate often points to regions where the industry either faces significant challenges or encounters notable opportunities.

Furthermore, the literature indicates discernible overlaps between certain topics and reveals the intertwined nature of these areas of focus. The confluence of ideas between the challenges of blockchain adoption (Topic 8) and blockchain-enabled traceability (Topic 5) implies that as enterprises integrate blockchain into their SCM processes, they invariably touch upon the pivotal aspect of enhancing traceability. This convergence is not surprising given blockchain's inherent capacity for transparent and immutable record-keeping. Similarly, the overlap observed between the challenges of blockchain adoption (Topic 8) and its impact on pricing strategies and decision-making (Topic 2) suggests a nuanced relationship. As organizations navigate the possibilities of blockchain adoption in SCM, they inevitably confront shifts in their conventional pricing strategies and decision-making paradigms. In sum, the multifaceted influence of blockchain on SCM is evident. The discourse underscores the recognition within the industry of blockchain's potential advantages but also emphasizes that its integration is riddled with complexities, necessitating an understanding of its repercussions on fundamental supply chain practices such as pricing and traceability.

Using PyLDAvis, a Python tool, as described by Sievert and Shirley in 2014 [68], researchers can understand the relative importance of the weights associated with the

selected topics in the LDA model related to blockchain research in SCM. Each pinpointed topic is visualized as a distinct colored circle on a two-dimensional plane, as shown in Figure 6. Among these, Topic 8 stands out as the most prominent circle, emphasizing its pivotal role in the current research environment. Although there are visible intersections among topics, Figure 6 still presents clear differences in the sizes of circles, indicating a broad spectrum of topics associated with blockchain in SCM. This representation highlights numerous specialized areas within the broader context of blockchain and SCM, each attracting specific attention. The breadth of these topics, juxtaposed with their observed overlaps, might signal uncharted territories for interdisciplinary inquiries. Thus, scholars are presented with the prospect of probing these interlinkages, potentially unearthing novel viewpoints and bolstering a comprehensive grasp of blockchain applications within diverse supply chain scenarios.

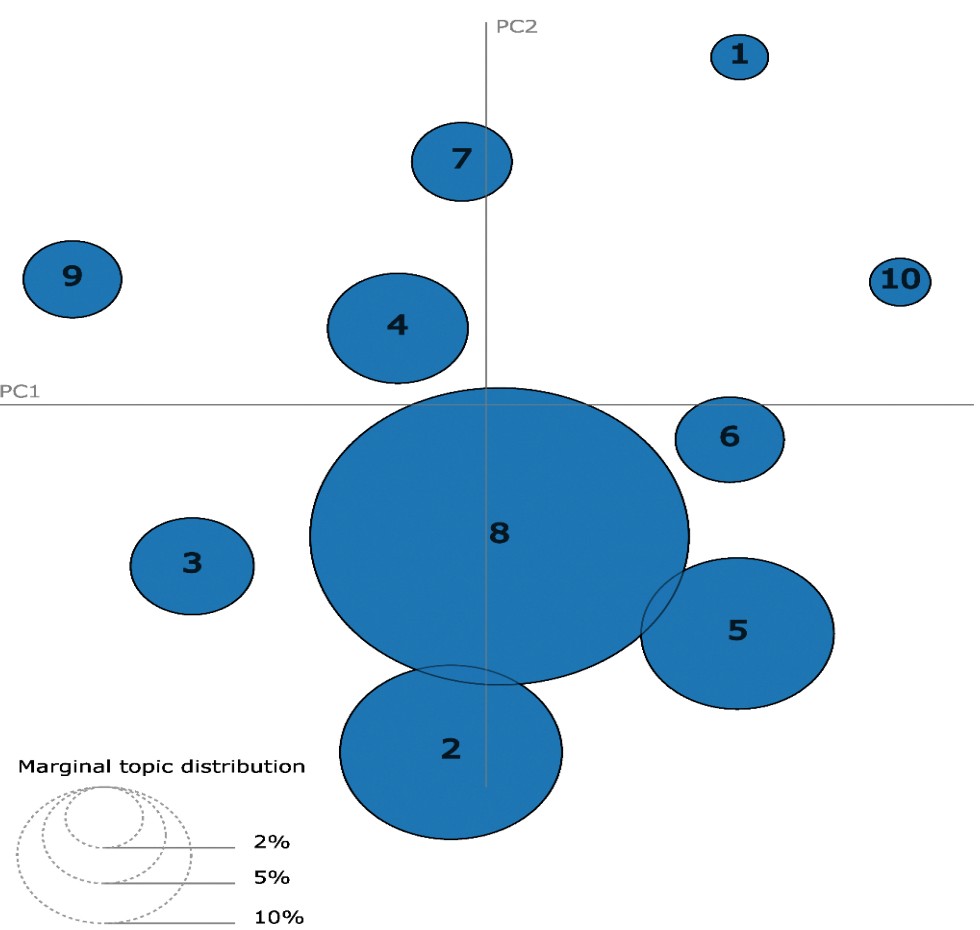

**Figure 6.** Intertopical distance map.

Within the scope of blockchain research in SCM, the data presented in Figure 7 unveil insights into the research's emphasis on various topics. From an impressive pool of 694 articles, Topic 8, which pertains to the challenges, adoption, and transformations related to blockchain in SCM, emerges as the predominant area of focus. Topic 2, which discusses blockchain's influence on pricing strategies and decision-making in SCM, is also of keen interest to researchers, evidenced by its 373 papers. Topic 5, centered on blockchain-enabled traceability, is also prominent, with 336 articles.

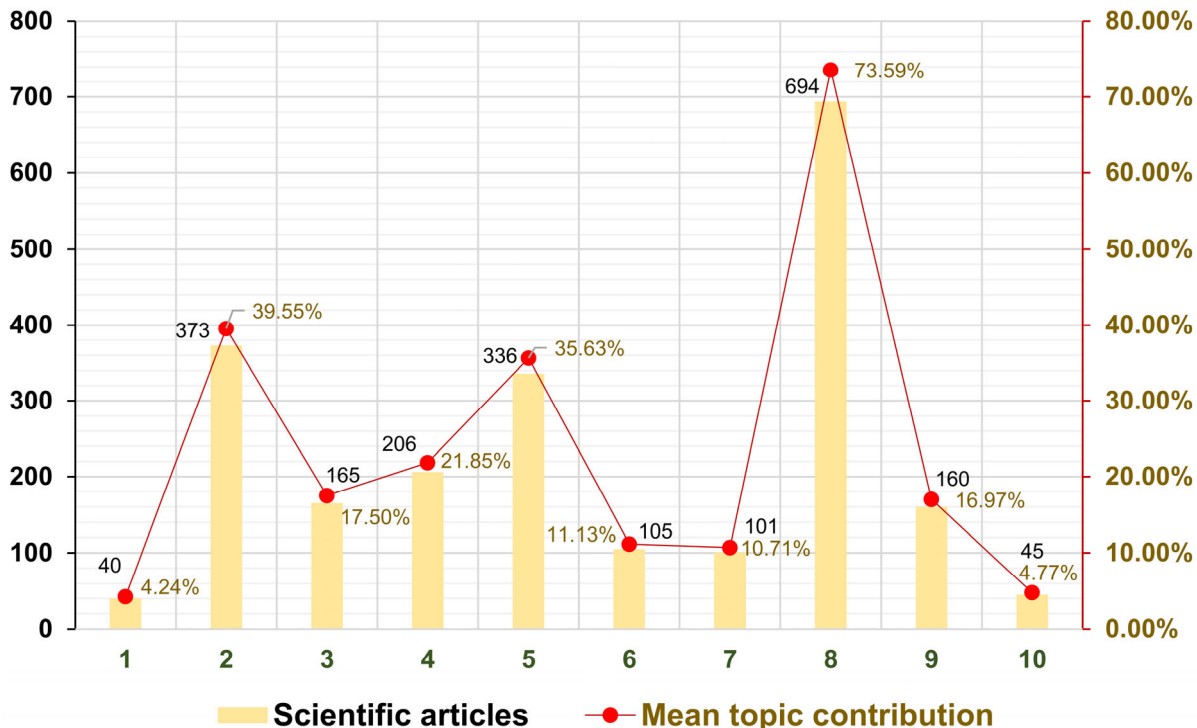

**Figure 7.** Topics distribution amongst documents.

In stark contrast, certain topics appear to be relatively less explored. For instance, Topic 1, delving into the application of blockchain for humanitarian supply chains, and Topic 10, emphasizing blockchain in the halal food supply chain, are represented by 40 and 45 papers, respectively. The median and average number of papers across these topics stand at 162.5 and 222.5, indicating that while some areas of blockchain in SCM receive considerable academic attention, others remain on the fringes of discourse. The pronounced interest in the challenges and transformations associated with blockchain adoption in SCM could potentially reflect the intricacies and hurdles that enterprises encounter in real-time integrations. Similarly, the noticeable number of articles discussing pricing strategies showcases the sector's need to understand the economic ramifications of blockchain adoption. Conversely, the fewer articles on the applications of blockchain in humanitarian contexts and halal food chains might indicate niche areas, suggesting significant potential for more granular investigations.

The overall distribution of research articles is distinctly skewed, underscoring the differential emphasis on various facets of blockchain's application in SCM. While some topics bask in the academic limelight, others are poised for deeper exploration and discovery. Multiple factors, ranging from the immediate applicability of topics and real-world challenges to the alignment with current global trends, could influence this distribution. This academic landscape presents a blend of established theories and nascent research opportunities, opening doors for both scholars and industry experts to further enrich the discourse on blockchain in SCM.

## 4. Discussion of Topics

### 4.1. Blockchain for Humanitarian Supply Chains

In the realm of blockchain research within SCM, an analysis of the LDA model underscores blockchain technology's promising role in augmenting coordination, streamlining information alignment, and proffering solutions tailored to humanitarian supply chains. The essence of this role is encapsulated by keywords such as "coordination", "humanitarian", "solution", and "emergency" (see Table 3), painting a vivid picture of blockchain's potential transformative influence.

**Table 3.** LDA model results.

| Topic | Keywords | Theme |
|:---:|:---:|:---:|
| 1 | 0.013*"HSC" + 0.010*"sharing" + 0.008*"coordination" + 0.007*"green" + 0.007*"driver" + 0.006*"knowledge" + 0.006*"humanitarian" + 0.006*"emergency" + 0.006*"solution" + 0.005*"commercial" | Blockchain for humanitarian supply chains |
| 2 | 0.052*"blockchain" + 0.036*"chain" + 0.034*"supply" + 0.019*"product" + 0.016*"information" + 0.014*"manufacturer" + 0.013*"retailer" + 0.013*"supplier" + 0.012*"cost" + 0.011*"model" | Blockchain's impact on pricing strategies and decision-making in SCM |
| 3 | 0.030*"blockchain" + 0.020*"financing" + 0.019*"finance" + 0.018*"chain" + 0.015*"supply" + 0.015*"SCF" + 0.012*"system" + 0.012*"financial" + 0.011*"smart" + 0.010*"model" | Blockchain's role in supply chain finance |
| 4 | 0.047*"food" + 0.023*"blockchain" + 0.018*"chain" + 0.017*"supply" + 0.014*"traceability" + 0.011*"intention" + 0.010*"system" + 0.010*"technology" + 0.010*"product" + 0.009*"adoption" | Blockchain in the food supply chain |
| 5 | 0.026*"chain" + 0.026*"blockchain" + 0.024*"supply" + 0.020*"data" + 0.017*"traceability" + 0.014*"system" + 0.012*"product" + 0.011*"transaction" + 0.010*"management" + 0.010*"SC" | Blockchain-enabled traceability |
| 6 | 0.031*"blockchain" + 0.014*"food" + 0.013*"analysis" + 0.012*"FSC" + 0.012*"technology" + 0.011*"system" + 0.010*"challenge" + 0.009*"application" + 0.008*"chain" + 0.007*"management" | Challenges of blockchain implementations in SCM |
| 7 | 0.018*"chain" + 0.017*"supply" + 0.016*"practice" + 0.015*"management" + 0.014*"sustainable" + 0.013*"blockchain" + 0.011*"performance" + 0.009*"barrier" + 0.007*"industry" + 0.007*"relationship" | Blockchain's impact on supply chain sustainability and performance |
| 8 | 0.061*"blockchain" + 0.043*"supply" + 0.042*"chain" + 0.014*"technology" + 0.012*"management" + 0.009*"adoption" + 0.007*"industry" + 0.007*"application" + 0.006*"data" + 0.006*"challenge" | Blockchain adoption, challenges, and transformations in SCM |
| 9 | 0.059*"logistics" + 0.019*"service" + 0.017*"e-commerce" + 0.017*"trade" + 0.016*"transport" + 0.010*"customer" + 0.010*"cross-border" + 0.009*"delivery" + 0.008*"firm" + 0.008*"data" | Blockchain in e-commerce operations |
| 10 | 0.036*"halal" + 0.028*"food" + 0.016*"information" + 0.008*"system" + 0.007*"traceability" + 0.007*"consumer" + 0.007*"quality" + 0.006*"blockchain" + 0.006*"risk" + 0.006*"institutional" | Blockchain in halal food supply chain |

Note (*): The weight or probability of the associated word in the given topic.

At the forefront of this exploration stands the study of Dubey et al. [69], which interrogates how blockchain technology can enhance coordination and refine information

alignment in humanitarian logistics. A complementary perspective is provided by the work of Dubey et al. [70], which envisages blockchain as an instrumental force driving operational transparency, instilling trust, and fostering collaboration in disaster-relief contexts. Deepening the conversation, Baharmand et al. [71] offer a nuanced understanding of the motivators and potential roadblocks associated with blockchain incorporation in humanitarian supply chains. On a related note, Ozdemir et al. [72] propound an in-depth exploration of how blockchain might counteract inherent challenges that plague humanitarian logistics. A synthesized overview is presented by Hunt et al. [73], weaving together myriad perspectives on the integration of blockchain in humanitarian operations.

Pivoting to the dimension of traceability in humanitarian logistics, Masudin et al. [74] delve into the multifaceted relationship between the adoption of technologies such as electronic data interchange (EDI), blockchain, and radio frequency identification (RFID) and the ripple effects on traceability and performance. Notably, while EDI's influence appears subdued, the role of blockchain and RFID emerges as crucial for enhancing trace-ability. Additionally, Chen et al. [75] introduce a pioneering approach to discern pivotal elements impacting blockchain's adoption. Using the avant-garde fuzzy large-scale group-DEMATEL methodology, the study highlights disintermediation as a primary factor, closely followed by anonymity and security, reaffirming blockchain's game-changing potential in humanitarian undertakings. Drawing the lens to barriers faced in blockchain adoption within humanitarian supply chains, Sahebi et al. [76] stand out. Employing a hybrid Fuzzy Delphi and Best-Worst method (BWM), the research identifies key challenges, with regulatory uncertainty, training deficits, and sustainability costs taking precedence.

In the realm of humanitarian operations, the pressing need for real-time data during emergencies makes the quick implementation of blockchain technology vital [77]. How-ever, challenges arise due to diverse operating standards among various humanitarian organizations and NGOs [73,76]. There is the matter of trust, particularly pertinent in high-stakes, crisis-driven environments where multiple actors must rely on a unified ledger [70]. Theoretical challenges revolve around coordinating disparate humanitarian efforts under a unified blockchain system, ensuring data privacy while maintaining transparency, and addressing the ethical considerations of data storage in crisis contexts [76]. On the practical side, areas hit by disasters might face infrastructural limitations, making the deployment of blockchain solutions logistically challenging [72]. Furthermore, training deficits are particularly pronounced, as introducing a new technology requires not only infrastructural adjustments but also capacity building among the ground staff [73,76]. In such resource-constrained settings, the economic costs of the transition to blockchain and its long-term sustainability, both environmentally and operationally, are of paramount concern [76,78].

The insights related to Topic 1 offer invaluable guidance for stakeholders aiming for efficient blockchain integration in humanitarian frameworks. In sum, these academic forays enrich our understanding of blockchain's multifaceted role in humanitarian supply chains. They simultaneously celebrate its potential while urging cognizance of challenges, outlining a balanced narrative of promise and prudence. Building upon this foundation, certain areas in the literature remain uncharted, presenting potential research gaps:

- Integration complexities: While blockchain's potential is emphasized, there is a dearth of literature exploring the practical complexities of integrating blockchain into existing humanitarian SCM systems.
- Cultural and geopolitical factors: The current research primarily focuses on technolog-ical challenges, leaving a gap in understanding how cultural and geopolitical nuances can influence blockchain adoption in different humanitarian contexts.
- Comprehensive cost-benefit analysis: Given the discussed advantages of blockchain, a deeper understanding is required concerning the challenges and upfront costs, providing a balanced view.
- Synergies with other technologies: Investigate the optimal combination of blockchain with other technologies, such as AI and big data analytics for enhanced traceability and performance in humanitarian SCM.

- Training needs: With identified training deficits, future research could delve into the specifics of these needs, leading to the creation of standardized training modules for blockchain adoption in humanitarian settings.

*4.2. Blockchain's Impact on Pricing Strategies and Decision-Making in SCM*

Interpreting topic 2 in light of the given studies, it becomes evident that blockchain's impact on pricing strategies and decision-making in SCM is multi-dimensional, presenting both opportunities and challenges. In this context, the research of Zhang et al. [79] shed light on how blockchain, as a shared and transparent database, can potentially attract more consumers due to the clarity of product information. However, there is a double-edged sword at play. On the one hand, there is an allure for retailers in offering transparency through blockchain; on the other, this technology brings a significant risk of privacy leakage. Strategically, the findings suggest that retailers face a delicate balance. Blockchain's adoption is not always the most beneficial approach, especially when consumer privacy concerns outweigh the benefits of information transparency. A particularly intriguing insight is the differential impact on established versus new entrants in the retail market. Established retailers can leverage blockchain when both privacy concerns and information transparency are at moderate levels due to the inherent trust they have already built with consumers, something that newer entrants might struggle with.

Next, Liu et al. [80] add another dimension, illustrating the practical implications of blockchain in sectors such as healthcare. In the intricate world of vaccine distribution, where safety and traceability are paramount, blockchain emerges as a beacon of operational efficiency. The research underscores that the technology's adoption enhances the overall profit, consumer surplus, and societal welfare of the vaccine supply chain, serving as a testament to blockchain's transformative potential. Diving deeper into the intricacies of SCM, Wu and Yu [81] and Tao et al. [82] both underscore the profound influence of blockchain on platform supply chains. By alleviating information asymmetry and reducing transaction costs, blockchain paves the way for better pricing strategies and market control. However, the latter study [82] also provides a cautionary note, highlighting that while blockchain can usher in many advantages, there are circumstances where it might lead suppliers to compromise on product quality, which could adversely affect consumers.

In the pharmaceutical industry, blockchain aids in transparent pricing strategies by providing a clear record of production costs, distribution expenses, and other overheads [83,84]. This transparency can lead to fair pricing, as all stakeholders can verify the cost structures [85]. Similarly, in the agri-food sector, blockchain enhances traceability from farm to retail point of sale. This traceability mechanism may extend to a Quick Response (QR) or other 2D barcode on products that allows consumers to scan using a mobile phone and verify the origin and other unobservable credence claims (e.g., organic, halal, kosher) made by the brand owner. Consequently, firms can set premium prices for desirable credence claims such as organic or ethically sourced products that can be digitally verified [86]. On the decision-making front, Rejeb et al. [17] illustrate how blockchain's tamper-proof nature helps stakeholders make informed choices by accessing authenticable historical data. In situations where raw material costs fluctuate, having a reliable record can aid in forecasting and making timely procurement decisions [87].

The importance of timing in the adoption of blockchain technology is brought forward in the study of Ji et al. [88]. As consumers become increasingly discerning about product traceability, traditional systems that are centralized and vulnerable to data manipulation falter. Blockchain stands out as the solution to this problem, but its adoption needs to be strategically timed. Early adopters, especially among manufacturers, are positioned to gain significant advantages in terms of profit. The paper of Li et al. [89] takes a financial turn, examining the burgeoning realm of cryptocurrency payments within supply chains. Interestingly, the study finds that while priority blockchain payment options might seem enticing, they may not always be profitable for service providers, especially when the trust rate for these priority services is low. Lastly, Wu et al. [90] offer a broader perspective on the

adoption strategies across multiple supply chains. Here, consumer traceability awareness and the cost-sharing dynamics of blockchain-based traceability between manufacturers and retailers play pivotal roles. Interestingly, high consumer traceability awareness does not automatically translate to blockchain adoption, suggesting that supply chains need a nuanced understanding of the myriad factors at play. Overall, blockchain's role in influencing pricing strategies and decision-making in SCM is profound, bringing a mix of transformative benefits and intricate challenges. Strategic foresight, consumer insights, and a nuanced understanding of both the technological and market dynamics will be crucial. However, amidst these revelations, there are areas that remain ripe for further exploration:

- Consumer-driven dynamics: While established retailers may benefit from moderate levels of privacy concerns and transparency, how might evolving consumer preferences and awareness change this balance in the future? This especially pertains to younger generations who might prioritize transparency over established brand trust.
- Detailed sectorial analysis: While healthcare, notably vaccine distribution, has been explored, how might other critical sectors, such as perishable goods or high-end luxury products, grapple with the challenges and opportunities blockchain presents?
- Ethical considerations in SCM: Tao et al. [82] hint at a potential compromise in product quality. This warrants a deeper investigation into the ethical ramifications of blockchain in SCM, especially in sectors where quality directly affects consumer well-being.

*4.3. Blockchain's Role in Supply Chain Finance*

The research landscape on the integration of blockchain technology in supply chain finance (SCF) reveals a diverse set of perspectives and methodologies. Blockchain's emerging role in SCF underscores its ability to address inherent challenges and introduce revolutionary mechanisms to transform traditional SCF systems.

Related to Topic 3, Li et al. [91] underscore the salient advantages of blockchain in inventory financing, especially for high-value financing businesses. It portrays the blockchain's ability to stabilize financing systems by discouraging collusion and promoting compliance among small and medium-sized enterprises (SMEs), supervisory entities, and banks. The study delves into the intrinsic ability of blockchain's smart contracts to detect and prevent unauthorized transactions and collusions, emphasizing its significance in reshaping SCF's trust dynamics. Complementing the above, the exploration of Ma et al. [92] elucidates the transformative potential of a decentralized approach. By using the blockchain's inherent tamper-evident and decentralized characteristics, the proposed model facilitates improved credit circulation from core enterprises to upstream and downstream SMEs.

Furthermore, Zheng et al. [93] address one of SCF's core challenges: the barriers to credit data. The research amplifies blockchain's capabilities to enhance transparency, security, and decentralization in credit information sharing, thereby optimizing the credit system of SCF. In a similar vein, the research of Dang et al. [94] innovatively amalgamates deep learning with blockchain. This synergetic approach not only predicts potential credit risks but also fortifies the credit reliability and trustworthiness of SCF systems.

The challenge of privacy in SCF systems is approached in the study of Wu et al. [95], which underscores the urgent need for privacy-centric solutions in blockchain-based SCF systems. By proposing a nuanced privacy protection architecture, this study emphasizes the importance of safeguarding sensitive financial data while maintaining the transparency and decentralization virtues of blockchain. Additionally, Kabir et al. [96] examine the adoption determinants of blockchain in SCF through an expanded UTAUT framework. By presenting empirical insights, this study underlines the prominent drivers influencing blockchain's integration into SCF systems, offering a comprehensive understanding of its uptake dynamics. Moreover, the intricate relation between blockchain technology and risk mitigation is further accentuated in the study of Sun et al. [97]. By offering insights into the

mechanisms through which blockchain can significantly reduce financing risks, the authors strengthen the argument for blockchain's indispensable role in ensuring SCF robustness.

Moreover, Chen et al. [98] offer a detailed exploration of how blockchain can revolutionize traditional SCF models. By providing a comprehensive architecture and dissecting the credit disassembly mechanisms enabled by blockchain, the authors furnish compelling insights into blockchain's transformative potential. Lastly, Mahmoudi et al. [99] extend the discussion into a niche domain of the construction industry. The pertinent challenges in SCF solutions are addressed, emphasizing the alignment between blockchain features and the sustainability imperatives of the construction sector. In conclusion, the nexus between blockchain technology and SCF unveils a plethora of opportunities and challenges. The current body of research accentuates blockchain's transformative potential in ensuring transparency, efficiency, and robustness in SCF systems, especially for SMEs. As SCF evolves in tandem with technological advancements, the integration of blockchain promises a paradigm shift, heralding a new era of trust, compliance, and innovation in SCF. Untapped research avenues that could shape future inquiries include:

- Interdisciplinary approaches: With the confluence of deep learning and blockchain seen in Dang et al. [94], what other interdisciplinary amalgamations could be explored? For instance, how might quantum computing or the IoT further optimize SCF when combined with blockchain?
- SME-centric blockchains: Given the emphasis on SMEs, is there potential for specialized blockchain networks tailor-made for SME requirements in SCF? How would these networks differ from generic SCF-focused blockchains?
- Consumer perception and behavior: While the technical and enterprise facets of blockchain integration in SCF are studied extensively, there remains a gap in understanding how consumers perceive and react to blockchain-backed SCF systems. This is especially pertinent given the increasing consumer demand for transparency in financial matters.
- International regulations and compliance: The global nature of many supply chains makes this an imperative. How might differing regulatory environments across countries influence, or even inhibit, the universal adoption of blockchain in SCF? What sort of cross-border blockchain frameworks might emerge to ensure seamless SCF operations?

### 4.4. Blockchain in the Food Supply Chain

Topic 4's output from the LDA model emphasizes the profound significance of blockchain in enhancing the traceability of the food supply chain. Terms such as 'food', 'blockchain', 'supply chain', and 'traceability' emerge prominently, underscoring the central theme of this topic.

In a detailed exploration of the societal impacts of blockchain technology, Mangla et al. [100] focus on the milk supply chains in Turkey. By leveraging systems theory and system dynamics, the researchers map the information flow within these supply chains to enhance traceability. The results highlight the transformative potential of blockchain. The integration of this technology not only fortifies the milk supply chain's transparency but also fosters social sustainability. This is manifested in various dimensions, such as local embedding, animal welfare, and food security. Further, blockchain's adoption promises a comprehensive tracking system, amplifying the cooperative's mission to attain the Sustainable Development Goals, particularly in terms of promoting health and ensuring food safety.

Consumer perceptions and behaviors form an essential facet of this discourse. A distinct study investigates consumer purchase intentions for blockchain traceable coffee [101]. Drawing upon the theory of planned behavior (TPB), the research [101] juxtaposes blockchain-based traceability against a more conventional traceability certification. Intriguingly, this study unearths that variables such as environmental protections and trust were instrumental in enhancing TPB's predictive power for blockchain traceable

coffee [101]. These findings carry significant implications for marketers, suggesting that raising consumer awareness about product traceability is imperative for the success of such blockchain-driven initiatives.

Delving further into consumer dynamics, other research underscores the potential of blockchain in signaling quality within the food supply chain [102]. Through a series of experiments, it is revealed that blockchain labels bolster consumers' perceptions of product quality. This elevated perception subsequently magnifies their purchase intentions. Interestingly, this impact is most pronounced for lesser-known brands, signifying blockchain's role as a pivotal tool for emerging brands aiming to create a niche in the market. This is likely at least a partial explanation for the rapid uptake of blockchain by supply chain newcomers.

On a regulatory front, blockchain's influence permeates the very architecture of food quality and safety standards. The European Food Safety and Quality System, a robust framework, recognizes traceability as a cornerstone. From a legal perspective, the deployment of blockchain can revolutionize this domain, ensuring impeccable traceability across the entire food production spectrum. The innovations in this field are not just limited to traceability but also accessibility. The FoodSQRBlock framework epitomizes this evolution [103]. By fusing blockchain technology with QR codes, this novel system ensures that both producers and consumers can effortlessly access, verify, and trace food production information, highlighting the potential of blockchain to revolutionize traditional agri-food systems.

Blockchain's adoption in retail food supply chains from a consumer standpoint has also been scrutinized. An extended technology adoption model (TAM) identified key drivers such as perceived security, privacy, and trust, which underpin consumers' inclination to use blockchain for tracing product origins [32]. This suggests that beyond traceability, blockchain also instills a sense of security and trust among consumers. When supply chain systems lack transparency, it may negatively impact consumer health and safety if the source of any unsafe food is unknown and a market-wide withdrawal or recall cannot be completed. Blockchain offers a robust solution to these challenges of traceability and information security. Lastly, the case of Lavazza, a renowned Italian coffee company, offers tangible insights into the real-world application of blockchain [104]. By integrating this technology, Lavazza catered to both internal stakeholders' need for enhanced traceability and consumers' demand for transparency. Such initiatives signal a trend where businesses actively harness blockchain to address evolving market demands. In conclusion, the studies underpinning Topic 4 illuminate the transformative potential of blockchain in the food supply chain. Whether it is enhancing traceability, instilling trust among consumers, or revolutionizing traditional systems, blockchain could help to reshape the food industry in the years to come. Related to Topic 4, several potential research trajectories and broader reflections emerge:

- Decentralization and ethical considerations: While blockchain's decentralized nature can amplify traceability, it might also introduce new ethical concerns. For instance, how might data ownership play out when every stakeholder has equal access to the information? Can there be instances where too much transparency becomes counterproductive?
- Local vs. global supply chains: Does blockchain's efficacy vary between local and global food supply chains? Are there unique challenges and benefits that manifest at different scales of operation?
- The role of cultural context: How do cultural attitudes towards food consumption, origin transparency and blockchain technology intersect? For instance, would blockchain-traceable products be equally valued across diverse cultures, or are there regional variations?

*4.5. Blockchain-Enabled Traceability*

The emergence of blockchain technology has introduced novel approaches to addressing the complexities and challenges inherent in SCM, particularly with regard to ensuring traceability, authenticity, and data security. Azevedo et al. [105] underscore the intricacies of global supply chains that span multiple borders and organizational interfaces. The authors propose a holistic approach that leverages the benefits of blockchain to provide traceability, chain of custody, and transparency. By integrating digital certificates with Supply Chain Actors (SCAs) and product identification, an Ethereum-based smart contract solution is proposed, emphasizing the significance of off-chain data storage solutions such as WalliD. This architectural design offers a promise of decentralized, trustful assurance of provenance and traceability, which was effectively demonstrated through its application in a real food supply chain use case.

Complementing this, Ho et al. [106] delve into the critical role of Aircraft Spare Parts Management (ASPM). Highlighting the International Air Transport Association's emphasis on the quality of traceability data, the study proposes a blockchain-based system that addresses the data quality challenges resulting from the complexities of multi-stage supply chains. By integrating Hyperledger Fabric and Hyperledger Composer, the system ensures data integrity, trustful data sharing, and improved traceability data quality. This research underscores the potential of blockchain to play an instrumental role in fostering the digital twin concept in the aviation industry.

In the agri-food sector, Xu et al. [107] present an innovative solution for addressing data traceability in the grain and oil industry, which is quintessential for ensuring quality and safety. The study outlines the significance of a cross-chain information interaction mechanism and proposes a model that safeguards traceability information, ensuring efficiency, credibility, and security. This contribution is pivotal as it addresses a niche yet crucial area in the food supply chain. The versatility of blockchain is further accentuated by Pérez et al. [108]. Focusing on the retail industry, this research elucidates how blockchain can be employed to monitor various facets of the apparel industry, from raw material sourcing to the final product. By proposing blockchain actors for each product stage, the paper lays the groundwork for ensuring transparency, authenticity, and reliability in the supply chain.

Extending the narrative to complex assembly structures, Kuhn et al. [109] offer a decentralized blockchain application named TokenTrail. The proposed solution uniquely addresses the traceability needs of multi-hierarchical assembly structures by developing an assembly token manager, which directly represents intricate assembly processes within a smart contract.

The interdisciplinary integration of blockchain with the IoT is also explored in the academic literature. For instance, Pincheira et al. [110] characterize the resource implications of integrating blockchain with low-cost IoT sensors, providing a cost model for blockchain infrastructure and thereby offering a comprehensive understanding of the economic feasibility of such integrative systems. Varavallo et al. [111] introduce a traceability platform with a reduced environmental footprint. Utilizing Algorand Blockchain's Pure Proof-of-Stake mechanism, the platform promises scalability, minimal energy consumption, and real-time data availability, applied specifically to the Fontina PDO cheese supply chain. Lastly, Fernando et al. [112] showcase the potential of blockchain to ensure traceability and control in drug distribution, emphasizing the characteristics of transparency, distribution, immutability, and peer-to-peer transactions.

In conclusion, across diverse sectors, from agriculture to aviation, from apparel to pharmaceuticals, blockchain-enabled traceability stands out as a transformative solution, promising enhanced transparency, reliability, and security in SCM. A closer examination of the literature and practical applications points to certain gaps and areas yet to be fully addressed:

- Implementation challenges: The literature provides a multitude of theoretical frameworks and models. However, there seems to be limited comprehensive analysis of

real-world implementation challenges, including factors such as stakeholder resistance, infrastructural limitations, and the training required for blockchain users.

- Interdisciplinary collaboration: While there is a recognition of the cooperation between blockchain and other technologies, such as IoT, in the study of Pincheira et al. [110], there is a relative lack of in-depth exploration into how these integrations can be smoothly operationalized and scaled across industries.
- Economic implications: The economic feasibility of blockchain integration, including cost-benefit analyses, ROI evaluations, and long-term financial implications for businesses of various scales, is not deeply explored.

### 4.6. Challenges of Blockchain Implementations in SCM

Blockchain technology has permeated various sectors. Topic 6 from the LDA model sheds light on the challenges faced during the implementation of blockchain in the supply chain. A deep dive into the studies provided offers a deeper understanding of this theme. For example, researchers examining blockchain challenges in the Indian healthcare sector stress the hurdles, such as limited expertise, technical issues, and resistance to change, with a significant pointer towards the lack of government initiatives [113]. Similarly, the exploration of Waste Electrical and Electronic Equipment (WEEE) management in developing nations signifies the barriers to data integrity, reluctance to adapt to change, and issues of information security [114]. These sentiments are mirrored in the agri-food supply chain study, which highlights challenges such as scalability, privacy concerns, and a general lack of regulations and training [115]. Further delving into the food sector, the thematic analysis stresses the nascent phase of blockchain adoption in food supply chains and highlights the associated challenges [116]. These range from technical to operational obstacles, mirroring the sentiments of traceability research, which emphasizes the importance of differentiating between public and sensitive data. This distinction is crucial to managing challenges pertaining to data access, storage, interoperability, and permanent operability.

Looking at multi-stakeholder perspectives in India, it is evident that complexity and compatibility with existing systems stand out as substantial barriers, emphasizing the critical role of top management support [117]. Another study, focused on the healthcare sustainable supply chain, identifies key challenges such as economic constraints, stakeholder commitment, data security threats, and poor infrastructure, underlining the significance of stakeholder involvement and supportive infrastructure [118]. Lastly, Kumar et al. [119] adopt a cautionary tone. While blockchain possesses the transformative potential to augment transparency and trust in SCM, it is essential to break through the hype. Blockchain, being a high-cost technology, should be employed judiciously, keeping in mind the associated costs and benefits. In summation, while the promise of blockchain technology in enhancing the robustness and transparency of supply chains is undeniable, it is vital to address the myriad challenges that arise in its wake. These span from technical, economic, and infrastructural issues to broader concerns about stakeholder commitment, government initiatives, and regulatory frameworks. The overarching sentiment from the studies suggests a cautious yet optimistic approach, emphasizing the importance of understanding, collaboration, and adaptability in navigating the complexities of blockchain adoption in SCM. Based on the analysis of Topic 6, the following are gaps for future research:

- Government initiatives: The evident lack of government initiatives in some sectors, especially in the Indian healthcare system, suggests a need for studies on the role of regulatory bodies in facilitating blockchain adoption.
- Data management in SCM: The distinction between public and sensitive data, along with concerns of data access, storage, interoperability, and permanent operability, indicates a need for comprehensive models and frameworks specifically tailored for SCM.
- Complexity and compatibility: With multi-stakeholder perspectives revealing issues of compatibility with existing systems, research focusing on bridging legacy systems with new blockchain solutions may be beneficial.

### 4.7. Blockchain's Impact on Supply Chain Sustainability and Performance

Topic 7 from the LDA output presents a complex interplay of factors, emphasizing the interaction of supply chains, sustainable practices, the role of blockchain, performance, barriers, and various industries, notably healthcare and fashion. Analyzing this topic in light of the studies provided reveals insights into the current trajectory of blockchain research in supply chain management.

Related to Topic 7, Umar et al. [120] illustrate the foundational role of blockchain in green SCM practices. Blockchain fosters information sharing, transparency, and collaboration on green issues. The terms "chain", "supply", "sustainable", and "blockchain" from the LDA output resonate with the study's findings, which emphasize the positive associations between blockchain-enhanced green manufacturing and both economic and social performance. Government support, as found in Umar et al. [120], further buttresses the incorporation of blockchain into green SCM, suggesting the important nexus between technology and governance. A different perspective is presented in Anonymous [121]. The fashion industry, which has its own set of complexities and challenges, seeks solutions in blockchain to foster transparency on a global scale.

Shifting geographical focus, Sundarakani et al. [122] delve into infrastructure readiness for sustainable practices in the transport sector. Interestingly, while the study found blockchain's adoption in the UAE to be nascent, the potential of blockchain to facilitate sustainable practices, especially in the energy sector, is evident. The Indian dairy industry, as explored in Kumar and Kumar [123], showcases a concrete example of how blockchain can transform a specific industry. It was observed that blockchain technology mediates the relationship between SCM practices and supply chain performance, suggesting blockchain's transformative potential. Diving into a specific application, Kleinknecht [124] studies the advantages of blockchain for meeting environmental regulations. Offering a more panoramic view, Ayan et al. [125] position blockchain as a pivotal tool across multiple industries. Their study elucidates the rise of blockchain-driven sustainable practices in diverse industries, further emphasizing blockchain's universality in enhancing supply chain efficiency. Finally, Prasad et al. [126] deep dive into the challenges of incorporating lean and sustainable manufacturing practices. Their research outlines economic, managerial, and organizational challenges but interestingly posits blockchain as a potential solution to overcome these barriers, especially in the micro, small, and medium enterprises (MSMEs) sector. In conclusion, the selected studies illuminate the multifaceted role of blockchain in shaping sustainable SCM across industries and regions. From enhancing green manufacturing practices and fostering transparency in the fashion industry to mediating supply chain performance and overcoming barriers in sustainable manufacturing, blockchain emerges as a transformative force in contemporary SCM research. The research gaps and areas that can be further explored include:

- Sector-specific sustainability metrics: While blockchain promotes sustainability across sectors, the exact metrics and benchmarks to measure 'sustainability' in each sector might differ. Research into defining and refining these metrics can add value.
- Evolution of green SCM practices: While blockchain is linked with enhanced green manufacturing, the trajectory and future stages of these green SCM practices, especially with the integration of newer technologies, need further exploration.
- Custom solutions for different business sizes: The challenges faced by micro, small, and medium enterprises (MSMEs) suggest a potential for blockchain solutions tailored to business size and capacity. Exploring how blockchain applications can be customized for different business scales is crucial.

### 4.8. Blockchain Adoption, Challenges, and Transformation in SCM

Blockchain adoption, challenges, and transformations in the supply chain, as delineated by the LDA topic modeling, pivots around key terms such as blockchain, supply chain, technology, management, adoption, industry, data, sustainability, and barriers. Analyzing the given studies vis-à-vis these themes allows for a cohesive understanding of

how research in this area is shaping up. Starting with Mangla et al. [127], the authors encapsulate the core of the topic—blockchain adoption in supply chain networks, especially with respect to food sustainability. The study delves into a specific case—the tea supply chain—providing a comprehensive framework that highlights the potential risks and barriers when merging blockchain with conventional methods. This aligns with the LDA theme that underlines the importance of understanding blockchain's potentialities and associated challenges.

In another research assessment, Rejeb and Rejeb [128], the focus is broadened to investigate the intersection of blockchain and supply chain sustainability. The paper's exploration of the triple bottom line framework—economic, social, and environmental dimensions—cements the LDA topic's emphasis on sustainability, management, and business. The paper effectively elucidates the transformative potential of blockchains, highlighting how they can foster new business models, cost advantages, and social equity. Moreover, van Hoek [129] offers an empirical vantage point on blockchain's adoption in the supply chain. The study, drawing from a workshop, delves into both the potential drivers for blockchain adoption (such as transparency and process improvements) and the barriers, especially concerning the cost-benefit understanding of this technology. Diving deeper into the sustainability aspect, Munir et al. [130] bring forward an exhaustive literature review on blockchain's potential to transform supply chains sustainably. Nazam et al. [131] add a fresh perspective. While it highlights the LDA's emphasis on adoption barriers, the study uniquely focuses on the textile industry, highlighting specific challenges faced during blockchain integration. Its rigorous approach in identifying and prioritizing barriers across technological and human resource dimensions further sharpens the focus on the topic's core—understanding the challenges in blockchain adoption for a sustainable supply chain.

Overall, these studies collectively elucidate the multifaceted nature of blockchain adoption in the supply chain, diving deep into its potential, challenges, and transformative capacities across diverse industries. Their synergistic focus on blockchain's integration barriers, sustainable practices, and real-world implications resonates robustly with the thematic core of the LDA model's Topic 8. The following research gaps need to be addressed in the future:

- Temporal evolution of blockchain adoption: While the current adoption challenges and benefits are explored, there is little insight into how these challenges and advantages might evolve over time. How will changing technological landscapes influence the dynamics of blockchain integration in the supply chains?
- Cross-industry comparative analysis: While individual sectors, such as the tea supply chain or the textile industry are explored, a direct comparative analysis between different sectors and their unique challenges or advantages in blockchain adoption might be lacking.
- SMEs vs. large corporations: Most studies might focus on larger supply chains or broad industry insights. How does blockchain adoption play out differently for SMEs compared to multinational corporations? What are the unique challenges or potentials for each?
- Cultural and geographical influences: How do cultural and regional factors influence the adoption of blockchain in supply chains? For instance, might the adoption barriers in the textile industry differ from one country or region to another?

### 4.9. Blockchain in E-Commerce Operations

The rapid ascent of blockchain in cross-border e-commerce and its applications in SCM has garnered considerable attention in recent academic research. The intersection of these fields primarily emphasizes the pivotal role of blockchain in enhancing the efficiency, traceability, and security of supply chain operations within the cross-border e-commerce context. A notable challenge in cross-border e-commerce is the efficient allocation of service capacity, especially in the sphere of third-party forwarding logistics (3PFL). This challenge emanates from the sporadic nature of order arrivals in such logistics operations.

In this regard, Ren et al. [132] elucidate this point by advocating for the use of deep learning approaches, particularly the Seq2Seq-based CNN-LSTM, to predict demand for logistics services. When juxtaposed with traditional models, the deep learning-based approach proves to be superior in both forecasting and capacity allocation. It showcases the value of integrating cutting-edge machine learning with logistics operations in e-commerce, particularly when dealing with unpredictable order dynamics.

The complexity of cross-border trade among Asian economic powerhouses such as Korea, China, and Japan gives rise to another fascinating area of study. By developing a blockchain-based cross-border e-commerce ecosystem, Shen et al. [133] posit that blockchain's core features, including decentralization and anti-counterfeiting traceability, can resolve existing bottlenecks. However, despite its promises, the technology is not without its setbacks. Issues such as low transaction concurrency and security loopholes need addressing for blockchain to truly revolutionize cross-border trade. Further building on the traceability facet of blockchain, Lee and Yeon [134] emphasize the importance of blockchain in preventing the proliferation of counterfeit goods. By enabling consumers to determine the authenticity of products and report counterfeits before making payments, blockchain plays a quintessential role in establishing trust and authenticity.

The competitive landscape of e-commerce is also affected by blockchain, especially when it comes to information disclosure. Sellers have the daunting task of accurately representing their products to gain consumer trust. Relatedly, Song et al. [135] portray blockchain as a tool that can mitigate information distortion and ensure efficient information transmission, particularly in competitive settings. Their study suggests that blockchain becomes indispensable for e-commerce sellers when either the consumer trust is low or the cost of blockchain implementation is feasible. Customs and trade facilitation represent another pivotal area where blockchain can bring transformation. Alqaryouti and Shaalan [136] put forth a framework to simplify cross-border e-commerce trade. By leveraging blockchain's inherent features, this framework seeks to instill trust among e-commerce stakeholders and streamline customs procedures, thus contributing to the larger e-commerce ecosystem.

An overarching view on the topic is provided by the study of Zhou and Liu [137], which underpins the vast applications of blockchain in the cross-border e-commerce supply chain, ranging from platform management to data governance. While the practical applications of blockchain in this sector are still budding, the research strongly hints at its impending ubiquity. The utility of blockchain in ensuring transparency and traceability is further highlighted in prior research on decentralized access to e-commerce products and the role of blockchain in sustainable B2B e-commerce supply chain finance [138,139]. Scholars accentuate blockchain's capacity to enhance both social and financial sustainability in the e-commerce landscape. In conclusion, blockchain's promise in the realm of cross-border e-commerce and SCM is palpable. Whether it is optimizing logistics, establishing trust, ensuring product authenticity, or facilitating smoother trade processes, blockchain's potential seems boundless. However, as with any burgeoning technology, challenges persist, and future research and collaborative efforts are necessary to harness its full potential. Amidst the detailed exploration of blockchain's role in cross-border e-commerce and SCM, numerous potential research avenues emerge, signaling untapped opportunities for scholarly inquiry:

- Behavioral response to blockchain adoption: With the rise of blockchain, how do consumers, retailers, and suppliers respond behaviorally to its adoption? Is there a measurable shift in trust or purchase behavior once consumers are aware of blockchain-backed authenticity checks?
- Economic impacts on SMEs: While large corporations might have the resources to integrate blockchain, how does this technology influence SMEs operating in cross-border e-commerce? Are the barriers to adoption different for them compared to industry giants?
- Integration with other emerging technologies: The integration of machine learning with blockchain, as mentioned, opens up the discussion on the potential synergies

between blockchain and other emerging technologies, such as IoT or augmented reality. How can these technologies converge in the e-commerce landscape?

- Regulatory and compliance challenges: With different countries having varying regulations for e-commerce, how does blockchain navigate this complex web? Are there instances where the decentralized nature of blockchain conflicts with stringent regulatory norms?

*4.10. Blockchain in Halal Food Supply Chain*

The integration of blockchain technology into SCM, especially in the halal food sector, has emerged as a central theme in both the LDA output and the studies. This becomes even more critical in nations such as Indonesia and Malaysia, where halal adherence is essential due to the significant Muslim populace.

The Indonesian emphasis on rigorous halal certification is evident in the study of Akbar et al. [140]. This research highlights an initiative aimed at bolstering the traceability of halal chicken products through blockchain. By adopting this method, consumers can gain increased assurance about the adherence to halal standards throughout the processing stages. Similarly, the halal beef sector also grapples with transparency issues. Hidayati et al. [141] illuminate these challenges, specifically in Medan City, where tracing the halal status of beef proves problematic due to opaque supply chains. The authors illustrate a transparent system where data remain not only accessible but also immutable, ensuring authenticity. Adoption of these systems is not solely predicated on technology; it is intertwined with sociocultural dynamics. Sumarliah et al. [142] examine these nuances by blending halal-focused attitude, innovation diffusion, and institutional theories. The study's findings offer a compelling insight: businesses deeply entrenched in halal-focused practices are more inclined to embrace blockchain-based halal traceability.

Moving from adoption to implementation, Ali et al. [143] discuss the rewards and challenges intrinsic to the integration of blockchain in the halal food supply chain. Their research proposes a pragmatic framework to mitigate challenges associated with blockchain implementation in this sector. The halal food industry is particularly sensitive due to religious, ethical, and quality assurances that are paramount for consumers [143]. Blockchain's promise of traceability and transparency can address concerns about food origin, processing methods, and certification verifications [144]. However, challenges manifest in various ways. Theoretically, there is the issue of how to design a blockchain system that accommodates the various stakeholders in the halal food chain, from producers to certifiers to distributors, ensuring that each party can validate halal authenticity at every stage [145]. Practically, integrating blockchain may face resistance from traditionalists within the industry, wary of technology's role in religious compliance [146]. There are also concerns about the global standardization of halal certification processes and how a decentralized ledger can uniformly address these. The real-world case studies highlight the potential of blockchain in ensuring halal compliance, tracking product origin, and preventing fraudulent halal certifications [143]. However, they also underscore the need for stakeholder collaboration, infrastructural investment, and public awareness campaigns to make blockchain's integration in the halal food supply chain truly transformative [147].

Another study [148] underscores Indonesia's ambition to amplify its presence in the global halal market. It emphasizes a technological shift imperative to bestow more transparent and detailed information throughout the halal assurance process. Bringing a touch of skepticism to the discourse, Hew et al. [146] delve into the motivations behind manufacturers' participation in a blockchain-based system. The study reveals that manufacturers' perceptions and evaluations of the system predominantly influence their adoption decisions. Lastly, Tan et al. [149] navigate the intricate landscape of global supply chains in Malaysia. Here, ensuring halal food integrity becomes an arduous task. The research introduces a trailblazing traceability framework underpinned by real-life blockchain implementations, melding halal processes with advanced technologies for a comprehensive traceability solution from farm to fork. In summation, the fusion of blockchain technology in the halal

supply chain manifests as an intersection of technological innovation, religious fidelity, and socio-economic dynamics. The consistent exploration in this arena underscores its burgeoning importance, hinting at a potential overhaul of halal certification and traceability in upcoming years.

As scholars delve into the nuances of blockchain's integration in the halal food supply chain, several underexplored avenues present themselves, signaling rich opportunities for future research:

- Halal traceability assurance: As blockchain's transparency is championed in the halal food sector, is there a noticeable increase in consumer confidence and assurance levels when purchasing blockchain-certified halal products? How does it change consumer preferences and purchasing habits?
- Localization vs. globalization: In the context of halal food, how does blockchain adoption impact local producers versus global exporters? Are there unique challenges faced by local producers in Muslim-majority nations when integrating blockchain for halal certification?
- Cultural acceptance and tech adoption: Given the religious implications associated with halal, how do traditional communities perceive and embrace technological solutions such as blockchain? Does the blend of religious fidelity with technological innovation lead to enhanced or hindered acceptance?
- Halal blockchain startups and innovation: As blockchain emerges as a potential game-changer for halal certification, are there new startups or business models emerging to capitalize on this intersection? What innovations are they bringing to the table?

## 5. Conclusions, Implications, and Limitations

Broadly, SCM continually grapples with inefficiencies and a lack of transparency. The expansion of global markets exacerbates these challenges. In response, blockchain technology emerges as a potential solution to revolutionize supply chain systems through its core traits of decentralization and information security. Despite blockchain's recognized promise, there is a gap in deep, comprehensive reviews of existing literature, especially using advanced methods such as topic modeling. Our bibliometric study highlights the burgeoning academic interest in the fusion of blockchain and SCM. The nascent nature of these publications underscores the urgency of this discourse. A salient observation is the synergistic interdisciplinary collaboration among researchers, which reflects the inherently cooperative functionality of blockchain technology. The scholarly trajectory spanning from 2017 to 2023 evidences the global escalation in recognizing the transformative potential of blockchain technology within SCM. Leading academic journals in the field, such as "Sustainability," underscore the multifaceted implications of this technological application.

Applying the LDA, we identified ten crucial topics with two key themes: blockchain's transformative impact on supply chain economics and the challenges of its integration into existing systems. The demand for traceability has amplified blockchain's role in the food supply chain, ensuring product visibility. Other significant areas include its integration into humanitarian efforts, enhancing e-commerce operations, and ensuring compliance in the halal food supply chain. Our findings also unveiled overlaps between topics, showcasing the complexities of blockchain's incorporation into supply chains. Visualization tools, such as PyLDAvis, showed the interconnectedness of these themes. The varied emphasis across topics suggests a vibrant research landscape, with areas such as humanitarian use and halal food chains ripe for deeper investigation. In summary, the intersection of blockchain and supply chain research offers a blend of well-established concepts and emerging opportunities, hinting at a more efficient, transparent future for SCM.

Building on this, the real contributions of our study extend beyond the mere identification of topics and themes. We present a structured understanding of how blockchain can redefine the traditional constructs of SCM, potentially ushering in paradigm shifts in how industries operate. Our approach casts light on both mainstream areas of interest and marginal paths, particularly in the context of humanitarian and halal supply chains,

thus paving the way for future dedicated research. By explicitly highlighting the intrinsic difficulties of integrating blockchain with current systems, our research offers invaluable insights to businesses, equipping them with knowledge of potential pitfalls and strategies for seamless integration. The utilization of visualization tools provides an intuitive understanding of the blockchain-SCM landscape, potentially guiding better decision-making and strategy formulation for both researchers and practitioners. The emphasis on traceability underscores our advocacy for an evolution towards a more transparent and trustworthy supply chain model, promising benefits for all stakeholders involved. In essence, our study does not just analyze; it charts a potential roadmap for the future of SCM in the age of blockchain, signifying a profound shift in both theory and practice.

### 5.1. Theoretical Implications

The amalgamation of blockchain and SCM undeniably possesses profound theoretical implications. Firstly, the study accentuates the transformative potential of blockchain within SCM, reshaping foundational theoretical concepts of transparency, trust, and efficiency. Blockchain's decentralized nature offers a paradigm shift from centralized supply chain systems, prompting scholars to re-evaluate traditional models of SCM.

The critical role of blockchain in economic dynamics, particularly in pricing strategies and decision-making within supply chains, serves as a testament to its capability to redefine market equilibria and value distribution mechanisms. As industries grapple with integration challenges, the theoretical models of change management and technological adoption within SCM come into the spotlight, necessitating a re-assessment in the context of disruptive technologies such as blockchain. Moreover, the evident emphasis on traceability and authenticity, especially in the food supply chain, adds another layer to the consumer trust theory. Blockchain does not just enhance traceability; it redefines it, establishing a new theoretical benchmark for consumer trust based on transparency and immutable provenance. The following directions for future research are suggested:

- Deepening the niche areas: Given the peripheral status of topics such as blockchain's role in humanitarian contexts and halal food chains, there is ample scope for in-depth exploration. Researchers could delve into the challenges, benefits, and strategies for integrating blockchain in these specific contexts, perhaps even devising specialized blockchain protocols tailored for these scenarios.
- Integration challenges and solutions: The integration of blockchain into existing supply chain structures seems to be a major point of contention. Future research could focus on identifying sector-specific challenges and propose actionable strategies or frameworks to facilitate smoother integration.
- Consumer perception and behavior: With the evident emphasis on traceability and the role of blockchain in ensuring product authenticity, understanding consumer perception towards blockchain-enabled products could be fertile ground for research. Are consumers willing to pay a premium for products verified by blockchain? What drives their trust in such products?
- Interdisciplinary studies: The interdisciplinary essence of blockchain applications in SCM, ranging from sustainability to economics, signals the possibility of collaborative studies. Combining insights from different academic lenses could lead to a more holistic understanding of blockchain's role and potential in SCM.
- Economic ramifications of blockchain in SCM: Given the importance of pricing strategies and decision-making, studies that delve deeper into the long-term economic implications of blockchain in SCM would be beneficial. How does blockchain influence cost structures, profit margins, and competitive advantage in the long run?
- Evolutionary perspective: Tracking the temporal evolution of blockchain applications in SCM could provide valuable insights. How have theoretical perspectives and practical applications changed over time, and what does this evolution signal for the future?

### 5.2. Practical Implications

The intersection of blockchain technology and SCM heralds a cascade of significant implications for businesses, policymakers, and consumers alike. The immutable and decentralized characteristics of blockchain promise to streamline administrative procedures, obviate manual verifications, and eliminate the necessity for intermediaries. This technological synergy provides businesses with a pathway for expediting operational processes, attenuating operational expenditures, and mitigating errors within supply chain functionalities. This streamlined approach fosters trust by providing digital assurances for product claims and traceability of products to their origin. Blockchain can play a critical role in verifying various product and process claims. Moreover, the decentralized nature of blockchain aids in forging supply chains that are not only transparent but also resilient against potential disruptions. This resilience is further heightened by blockchain's transformative effect on pricing strategies. With an open ledger system bringing forth unparalleled transparency, businesses will find themselves at an inflection point—either justifying their cost structures more coherently or leveraging this newfound transparency to command price premiums on authentic, verifiable products.

Amid these paradigmatic shifts, the intrinsic collaborative ethos of blockchain technology warrants emphatic emphasis. It catalyzes the emergence of cooperative ecosystems, wherein businesses—including potential competitors—may selectively share data strata to enhance efficiency and transparency across the industry. Such synergy can manifest in the real-world as industry consortiums or alliances, unified in their intent to harness blockchain's full potential. On the regulatory front, blockchain's promise lies in simplifying compliance and auditing processes. The technology's tamper-proof nature means that regulatory agencies can swiftly verify records, ensuring businesses adhere to industry norms and standards with greater ease. This streamlined compliance extends to humanitarian endeavors as well. In times of crises, blockchain's role becomes pivotal in ensuring efficient, transparent, and timely aid distribution, minimizing bureaucratic hurdles, and maximizing impact. For sectors that often remain overshadowed in global supply chains, such as agriculture or artisanal production, blockchain emerges as an empowering tool. It offers these sectors a mechanism to validate their products' authenticity, ensuring they get fair market recognition and compensation. Concurrently, as consumers become increasingly savvy about blockchain's capabilities, their demands for transparency and accountability will burgeon, exerting further pressure on brands to actively pursue blockchain integrations.

### 5.3. Research Limitations

This review has certain limitations that must be acknowledged. First and foremost, our analysis was based on the study of abstracts rather than full papers. This decision was taken to efficiently filter the vast amount of available literature, but it does come with the limitation of potentially missing out on the in-depth discussions and nuanced insights present in the complete papers. Additionally, our focus was primarily on journal articles, which means that we may have inadvertently overlooked valuable insights from conference proceedings, books, chapters, industry reports, and practical experiences of blockchain and supply chain professionals. It is essential to recognize that our dataset might have inherent biases due to our choice of specific databases and search criteria. We may not have captured every pertinent publication, especially those in languages other than English or those in less accessible databases.

These sources, particularly on-ground narratives from professionals, could offer practical insights that might not always be captured in academic writings. Moreover, despite being advanced, the LDA approach is fundamentally algorithmic. This means it identifies themes based on term frequencies, which can sometimes miss the depth or intricate contexts of specific discussions. Given the swift advancements in blockchain technology and evolving supply chain strategies, there is a potential that recent developments, post our cut-off date, have not been covered. Furthermore, the timeframe of our analysis, spanning up to July 2023, means that we are presenting a snapshot of the literature up to that point.

Moreover, our review's inclination towards major journals might introduce a certain bias. This could favor more mainstream perspectives and possibly sideline some lesser known yet equally significant insights. While we noted thematic overlaps in our review, a more detailed exploration of their intricate relationships and interconnections may be warranted in future studies.

**Author Contributions:** Conceptualization, A.R.; methodology, K.R.; software, K.R.; validation, A.R., J.G.K. and S.S.; formal analysis, A.R.; investigation, A.R.; resources, K.R.; data curation, K.R.; writing—original draft preparation, A.R.; writing—review and editing, J.G.K.; visualization, K.R.; supervision, S.S.; project administration, A.R.; funding acquisition, K.R. All authors have read and agreed to the published version of the manuscript.

**Funding:** This research received no external funding.

**Data Availability Statement:** Data are available upon request.

**Conflicts of Interest:** The authors declare no conflict of interest.

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
