# Peer review of "Exploring Blockchain Research in Supply Chain Management: A Latent Dirichlet Allocation-Driven Systematic Review"

_information, doi:10.3390/info14100557_

Round 1

Reviewer 1 Report

I thank the authors for reviewing their manuscript. The study is interesting because it explores the current state of implementation of blockchain technology in supply chain management. The study needs some substantial changes to be published.

In the introduction, clarify the objective of the study by highlighting a research gap. There are numerous systematic literature reviews related to blockchain and supply chain management (i.e. Varriale et al. 2020; Wang et al. 2019; Hastig et al., 2020). What differentiates your systematic literature review from others? In the introduction, it is necessary to clarify the methodology applied and the main contributions of the study.

It is necessary to clarify the research time horizon of the selected papers.

Insert the flow chart in the appendix in the methodology section.

The discussions are a mere literature analysis and already known elements are presented that do not add any novelty or originality to the manuscript. The description of numerous examples makes the discussion fragmentary and the ultimate purpose of this analysis is not understood. It would be more useful to comment on the data provided by the LDA in a more general and less punctual manner (i.e. by citing the numerous articles). Your study aims to provide an exploration of the state of the art of blockchain in supply chain management by clarifying the main research topics and identifying potential research gaps to be filled.

The 10 patterns identified are prone to overlap. For example, the topics of blockchain in food supply chains, blockchain in halal food supply chains and the topics of sustainability and tracking are often repeated and overlapping. These 10 models should have their own specific characterisation and avoid the same articles potentially being used in different topics, otherwise this classification is useless. 

In discussions, it is necessary to be direct and clear about the real contribution of the study.

Please do not use the term REF when citing studies (i.e. REF [7]). Rather edit the text to make it easier to read.

It is crucial to improve the English language, proofreading by a native speaker is absolutely necessary.

Varriale, V.; Cammarano, A.; Michelino, F.; Caputo, M. The Unknown Potential of Blockchain for Sustainable Supply Chains. Sustainability 2020, 12, 9400.

Wang, Y.; Han, J.H.; Beynon-Davies, P. Understanding blockchain technology for future supply chains: A systematic literature review and research agenda. Supply Chain Manag. 2019, 24, 62–84.

Hastig, G.M.; Sodhi, M.M.S. Blockchain for Supply Chain Traceability: Business Requirements and Critical Success Factors. Prod. Oper. Manag. 2020, 29, 935–954

It is crucial to improve the English language, proofreading by a native speaker is absolutely necessary.

Author Response

Dear Reviewer, 

Please find our answers to your comments in the attached file. 

Kind regards, 

The authors 

Reviewer 2 Report

Overall the paper is well written and analysed. There are 2 main concerns that need to be fixed as well as 1 major issue that, unfortunately, makes the validity of the paper questionable.

1) The introduction section does not clarify what sets this research apart from similar ones. The authors only state that compared to ref. 42,  their study will be grounded on a more expansive sample and will center its analysis on article abstracts. The fact that the authors study a wider amount of papers is positive but the fact that they only analyse them based on their abstract lowers the soundness of the findings. 

2) It is mentioned that traceability is one of the main applications of blockchain in supply chain management. However, the authors present very few examples. A paragraph should be added in the introduction section presenting valuable research. Suggested papers to be added are: 

Gayialis, S. P., Kechagias, E. P., Papadopoulos, G. A., & Panayiotou, N. A. (2022). A Business Process Reference Model for the Development of a Wine Traceability System. Sustainability14(18), 11687.

Omar, I. A., Debe, M., Jayaraman, R., Salah, K., Omar, M., & Arshad, J. (2022). Blockchain-based supply chain traceability for COVID-19 personal protective equipment. Computers & Industrial Engineering167, 107995.

Khanna, A., Jain, S., Burgio, A., Bolshev, V., & Panchenko, V. (2022). Blockchain-enabled supply chain platform for Indian dairy industry: safety and traceability. Foods11(17), 2716.

Kechagias, E. P., Gayialis, S. P., Papadopoulos, G. A., & Papoutsis, G. (2023). An Ethereum-Based Distributed Application for Enhancing Food Supply Chain Traceability. Foods12(6), 1220.

Wang, L., He, Y., & Wu, Z. (2022). Design of a blockchain-enabled traceability system framework for food supply chains. Foods11(5), 744.

3) The main issue that makes the methodology followed to perform the literature review is questionable as when applying the specific search term in the SCOPUS library, 5599 papers are found. This is about five times larger than the amount stated by the authors (1308). Therefore, a huge amount of valuable papers is unjustifiably excluded from this research, and its results definitely don't accurately present the relevant research. Also, as already stated, the authors only study abstracts and not full papers. All these limitations severely lower the robustness of the literature review. The authors need to add this to their limitations section and clearly explain why they have excluded so many relevant papers from their research. 

The use of English is very problematic. There are many grammar and syntax errors and sentences that make no sense. The authors need to proofread the paper and fix all the issues with the use of English. 

Author Response

(The authors gave the same response as above.)

Reviewer 3 Report

Some of the comments that can be addressed are as follows:

Introduction Clarity: While your introduction effectively sets the stage, consider providing more context on the specific challenges faced by supply chain management, such as notable examples of inefficiencies or disruptions, to engage the reader more deeply.

Research Objectives: Explicitly state the research objectives at the beginning of the paper. What are you trying to achieve with this comprehensive review?

Data Source and Methodology: Detail the source of your dataset, including how you collected and curated the 943 articles. Explain the rationale behind choosing the Latent Dirichlet Allocation (LDA) method for analysis. Provide the LDA model's parameters, such as the number of topics.

Thematic Structure: In the section discussing the thematic structure, provide a brief description of each of the ten central topics identified by the LDA analysis. Readers should gain a basic understanding of each topic without needing to refer to the entire dataset.

Discussion Elaboration: Expand on the discussions related to blockchain's transformative role in various supply chain aspects. Include practical examples or case studies where applicable to illustrate the impact.

Visualization Interpretation: When discussing the visualization tools like PyLDAvis, provide explanations of what the visualizations represent. Explain how they enhance understanding of the thematic structure and the interconnectedness of topics.

Challenges and Limitations: Go into more detail about the challenges and transformations of blockchain integration in supply chains, including specific challenges faced by industries like halal food supply chains or humanitarian contexts. Discuss these challenges from both a theoretical and practical standpoint. You can also discuss few papers in this direction such as: Internet of Things based Blockchain for Temperature Monitoring and Counterfeit Pharmaceutical Prevention

Impact on Pricing Strategies and Decision-Making: Provide concrete examples or models that demonstrate how blockchain technology impacts pricing strategies and decision-making in supply chain contexts.

Acknowledgment of Limitations: Acknowledge the limitations more explicitly. For example, mention the potential sources of bias in your dataset or the timeframe during which the analysis was conducted.

Future Research Directions: Offer more specific suggestions for future research. How can researchers build upon the topics you've identified? Are there any emerging trends or technologies related to blockchain and supply chain management that deserve attention?

Conclusion Recap: In the conclusion, summarize the key findings and contributions of your research concisely. Emphasize the practical implications and the value of your study to academics and industry practitioners.

Ensure that the language used is clear, concise, and free from unnecessary jargon. Aim for readability and accessibility for a broader audience.

Author Response

(The authors gave the same response as above.)

Round 2

Reviewer 1 Report

The authors have made an enormous effort by fulfilling all the required corrections. The paper impresses me with its clarity and linearity. The authors have provided a rich research agenda useful for academics. The paper is ready for publication.

Minor editing of English language required.

Author Response

Dear Reviewer,

Thank you immensely for your positive feedback and recognition of our efforts. We are delighted to hear that the paper resonates with its intended clarity. We are grateful for your guidance and constructive feedback throughout the review process.

Kind regards, 
The authors 

Reviewer 2 Report

The authors made the necessary improvements however, the explanation provided for the severe limitation of investigated papers isn't sufficient as in my opinion both the methodology and the small amount of papars limit the robustness and accuracy of the research. I leave it to the editor to decide if the research should be expanded. 

The quality has improved. Some minor changes need to be made concerning syntax and grammar issues. 

Author Response

Dear Reviewer,

Thank you for your feedback. We would like to clarify that our study analyzed a total of 943 papers, a comprehensive number that is consistent with the requirements for LDA-based systematic literature reviews. The methodology we employed is robust, with the identification of topics primarily relying on the abstracts of these 943 papers, ensuring a wide coverage of the existing literature. We believe this extensive dataset underpins the robustness and accuracy of our findings.

Furthermore, we have conscientiously acknowledged the limitations of our approach in the conclusion section to ensure transparency. For a better understanding of the appropriateness of our method, we recommend referring to the following seminal articles on LDA-based systematic literature reviews which have similarly relied on vast numbers of abstracts for topic modeling: 
https://www.sciencedirect.com/science/article/pii/S0968090X17300207
https://www.mdpi.com/2071-1050/13/3/1269
https://www.sciencedirect.com/science/article/pii/S0040162521008787#bib0116

Once again, we appreciate your insights and hope that this clarification addresses your concerns.

Kind regards, 
The authors 

Reviewer 3 Report

Authors updated the paper.

no coments

Author Response

Dear Reviewer,

We are grateful for your guidance and constructive feedback throughout the review process.

Kind regards, 
The authors